# Medically Tailored Grocery Deliveries to Improve Food Security and Hypertension in Underserved Groups: A Student-Run Pilot Randomized Controlled Trial

**DOI:** 10.3390/healthcare13030253

**Published:** 2025-01-27

**Authors:** Elaijah R. Lapay, Trevor M. Sytsma, Haley M. Hutchinson, Elliot J. Yoon, Scott A. Brummel, Linda Y. Tang, Elena G. Suarez, Kishen Mitra, Ryan M. Kane, J. Patrick Hemming

**Affiliations:** 1Department of Community Health, El Centro Hispano, Durham, NC 27707, USA; 2Root Causes, Duke University School of Medicine, Durham, NC 27710, USA; trevor.sytsma@duke.edu (T.M.S.); haley.hutchinson@duke.edu (H.M.H.); elliot.yoon@duke.edu (E.J.Y.); scott.brummel@duke.edu (S.A.B.); lindatangfr@gmail.com (L.Y.T.); elena.suarez@duke.edu (E.G.S.); kishen.mitra@duke.edu (K.M.); ryan.kane@duke.edu (R.M.K.); patrick.hemming@duke.edu (J.P.H.); 3Duke Office of Community Health, Durham, NC 27708, USA; 4Division of General Internal Medicine, Department of Medicine, Duke University, Durham, NC 27708, USA; 5Clinical and Translational Science Institute, Duke University, Durham, NC 27708, USA

**Keywords:** food security, hypertension, food assistance, minority health, community-based participatory research

## Abstract

Background/Objectives: Randomized controlled trials (RCTs) are needed to evaluate the impact of food is medicine (FIM) programs, such as medically tailored groceries (MTGs) to treat hypertension among diverse populations. Partnerships between academic centers’ student-run organizations (SROs) and community-based organizations (CBOs) offer critical safety nets for historically underserved groups, positioning these organizations to effectively undertake FIM programs among populations disproportionately affected by hypertension. We conducted an unblinded pilot RCT whose objectives were to assess the feasibility and acceptability of an SRO-coordinated, CBO-partnered MTGs intervention targeting blood pressure (BP) and food insecurity (FI) in underserved groups. Methods: Adult Black/African American and Hispanic/Latinx patients in Durham, North Carolina, where essential hypertension and FI were randomized (parallel arm, computerized 1:1 ratio) to 12 weeks of home-delivered, hypertension-focused MTGs plus in-person nutrition education sessions with compensation (intervention) versus data collection sessions with compensation (control). We offered transportation, childcare, and home visits to facilitate session attendance. The primary outcomes were the eligibility, enrollment, and retention rates (feasibility), and the survey feedback from the participants and CBO partners (acceptability). The secondary outcomes included the changes in the mean BP and median FI score with associated 95% confidence intervals. Results: Medical record screening identified 1577 eligible participants. Of the 94 reached to confirm eligibility, 77 met the enrollment criteria, and 50 were randomized (82% post-screen eligibility, 65% enrollment). A conventional content analysis of 15 participant surveys and CBO partner feedback affirmed the acceptability, noting intervention components that enhanced the retention (e.g., home delivery, transportation support, home visits). Pre–post analyses of secondary outcomes for 13/25 intervention and 15/25 control participants completing ≥2 sessions ≥2 months apart were performed. The intervention was associated with an average change in systolic BP of −14.2 mmHg (−27.5, −4.5) versus −3.5 mmHg (−11.7, 5.9) in the control group. The FI scores improved by −2 (−2.2, −0.5) in the intervention group and −1 (−1.3, −0.2) in the control group. No adverse events were reported. Conclusions: SRO-CBO partnerships could be feasible and acceptable avenues for conducting FIM trials among underserved populations. This multi-component FIM approach enhanced the study equity by addressing the participants’ disease-related social needs and warrants expansion into a powered RCT.

## 1. Introduction

Food insecurity (FI) and excess weight directly impact hypertension and incident cardiovascular disease [1]. Food is medicine (FIM) interventions show promise in improving health outcomes by integrating food and nutrition into healthcare systems to enhance patients’ access to nutritious foods [2,3]. Under new Joint Commission requirements, hospitals have recently increased attention to health-related social needs (HRSNs) screening and intervention—including FIM services—to reduce health disparities [4,5]. Health systems have been framed as a logical connection point for FIM interventions, but intersectoral partnerships aligning health systems with local governments, food systems, and community-based organizations (CBOs) to orchestrate such interventions are nascent and require further exploration [6,7,8]. Recent research has highlighted initiatives that integrate food and nutrition into healthcare and insurance payment models for the prevention, management, and treatment of diet-related diseases, including hypertension and diabetes [2,9]. These FIM strategies include produce prescription programs, medically tailored groceries (MTGs), and medically tailored meals. Each of these strategies have the dual aim of improving health outcomes and promoting health equity [6,9].

Recent FIM interventions demonstrated success in treating diet-related chronic diseases, like hypertension and diabetes [2,9,10]. However, the outcome effects are particularly variable for interventions using MTGs—the distribution of unprepared foods that recipients are meant to prepare themselves for consumption. While several randomized controlled trials (RCTs) have demonstrated MTGs’ ability to improve FI [11] and dietary intake/quality [11,12], only a small number of trials examined their impact on glucose control [11,12] and weight [12,13]. Notably, a 2024 Aspen Institute Report highlighted the lack of RCTs that assessed the impact of MTGs on blood pressure (BP) control, pointing to only a few single-arm studies that showed either a beneficial impact or null findings [2,14,15].

Historically underserved populations, such as Black/African American (B/AA) and Hispanic/Latinx (H/L) groups, experience FI and diet-related diseases, like hypertension, at higher, disparate rates compared with other populations [1,2]. Accordingly, groups like the American Heart Association (AHA) and the Aspen Institute have emphasized that FIM research should prioritize the inclusion of diverse populations, particularly those who have been historically and persistently underrepresented in biomedical research, including extant FIM trials [2,3,16]. The pursuit of equity-centered FIM research requires the intentional seeking and inclusion of eligible, diverse, and representative community members [2]. Community-based participatory research offers an approach to equitably engage communities of color in research, emphasizing the authority and credibility of local CBOs and coalitions. Accordingly, nutrition and lifestyle interventions to address hypertension that involve CBOs with racially and ethnically concordant representatives have demonstrated success with B/AA and H/L communities [17,18].

Student-run organizations (SROs) connected with academic medical centers—including student-run free clinics—act as critical safety nets for the U.S. medical system. These SROs often provide free primary medical care and occasionally integrate nutritional support programs into their services for underserved populations who might otherwise seek medical care in emergency and urgent care settings [19]. Generally, SROs fill two vital functions: providing health care to the uninsured and educating medical trainees to care for that population through community-engaged learning [19]. Because members of the B/AA and H/L communities are generally overrepresented in the uninsured population and disproportionately experience FI and diet-related chronic disease [19,20], SROs are potentially important sites to institute and study FIM interventions. Nearly all student-run FIM interventions have arisen from SROs, who engage in food security screening and respond to patient FI by either referring to external CBOs [21,22,23] or by integrating with CBOs to provide food items in clinic [24,25]. While SROs have demonstrated that they can successfully connect patients with resources to reduce FI, there are few reports of student-run FIM interventions and no reported examples of student-run FIM interventions measuring efficacy in treating or reversing diet-related chronic disease in historically underserved populations.

The Fresh Produce Program (FPP) at Duke University School of Medicine is one example of an SRO associated with an academic medical center in Durham, North Carolina. The FPP was started in 2017 by medical students as an in-clinic produce pantry serving outpatients experiencing FI [26]. In order to address accessibility during COVID-19, the FPP transitioned to a direct-to-home delivery model reliant on student and community volunteers to deliver MTG items to participants [26]. Between 2020 and 2024, the FPP expanded its services to over 600 families [26]. Participants in the program are 90% B/AA and/or H/L, reflecting the populations disproportionately experiencing dual food and transportation insecurity in Durham County [27]. Every two weeks, participants receive a 10–15-pound bag of MTG items, including free, locally sourced, fresh produce and some shelf-stable items, as well as community resource and nutrition education flyers with relevant recipes [26]. To date, the FPP has completed 250 consecutive weekly distributions of MTGs to community members [26]. However, its impact on diet-related chronic disease management has not been evaluated quantitatively. Therefore, we conducted a pilot RCT that aligned the FPP with local CBOs to deliver an intersectoral MTGs intervention that targeted BP and FI in underserved groups. To prepare for a powered RCT, our objectives were to assess the feasibility and acceptability of the intervention from an equity-focused lens [28] and thereby establish suitable procedures for the FPP and its partners to conduct study activities and measure outcomes.

## 2. Materials and Methods

We used the 2024 Aspen Food is Medicine Action Plan and the CONSORT 2010 extension to randomized pilot and feasibility trials to guide the analysis and reporting of this pilot RCT [2,29].

### 2.1. Trial Design

This was an unblinded, parallel-arm pilot RCT with 1:1 allocation assessing the feasibility and acceptability of an SRO-coordinated, CBO-partnered MTGs intervention that targeted BP and FI in B/AA and H/L participants that were experiencing hypertension and FI in Durham, North Carolina. The study activities occurred over 12 weeks between April and June 2023. The Duke Clinical and Translational Science Institute’s Community Engaged Research Initiative provided grant funding to support the work of a clinical research coordinator (CRC) and community health workers (CHWs) from a Latinx-serving CBO (El Centro Hispano). The study activities were conducted by volunteer undergraduate and medical students from the SRO (the FPP) and dietetics students from North Carolina Central University (NCCU)—all of whom completed mandatory institutional research ethics trainings—with the support of two faculty advisors from the Duke University School of Medicine and NCCU. The trial was approved by the Duke Health Institutional Review Board in Durham, NC (Pro00111072), and registered on clinicaltrials.gov (ID: NCT05926193). All the study activities were offered in both English and Spanish in accordance with the participants’ language preferences. Notably, this study was originally planned for 24 weeks between January and June 2023 but was shortened to 12 weeks between April and June 2023 due to the extended timeline of regulatory approval for CHWs and constraints on the availability of students to coordinate study activities outside of the academic year.

### 2.2. Participants

We queried the electronic health record (EHR, Epic, Madison, WI, USA) of Duke Health using Slicer Dicer (V5.2), a self-service reporting tool, to identify potentially eligible patients that met the following prescreening criteria: over 18 years of age, documented diagnosis of primary hypertension, residence in Durham County, documented FI, primary language of English or Spanish, and self-identification as either B/AA or H/L [30]. We limited the inclusion to individuals that identified as either B/AA or H/L to focus exclusively on Durham County’s two predominant historically underserved groups [27]. Patients with hypertension due to secondary causes (i.e., renovascular hypertension, hyperthyroidism, Cushing’s syndrome) were excluded. The initial query yielded 1577 patient records. A sample size of 100 was determined to be a reasonable initial recruitment target, representing over 5% of the patients that met the prescreening criteria and in alignment with the funding and personnel capacity to coordinate a study that lasted 12 weeks. Patient contact information and prescreening data were queried and extracted from the EHR using the Duke Enterprise Data Unified Content Explorer (DEDUCE) (V1.0) tool to maintain patient privacy and stored in a secure, cloud-based database (REDCap) built and maintained by the study team [31,32].

The patients that met the prescreening criteria were contacted by phone to confirm their information and verify their status as food insecure using the 2-question, Hunger Vital Signs questionnaire [16]. Patients were excluded if they did not screen as food insecure, were unable or unwilling to prepare or cook food at home, were experiencing homelessness, living or planning to move outside Durham County in the next twelve weeks, or pregnant/planning to become pregnant in the next twelve weeks. Eligible participants were invited to enroll in this study, and informed consent was obtained over the phone and documented through REDCap (V6.5.15) [32].

### 2.3. Randomization

Following the receipt of informed consent, the participants were randomized in block sizes of four into the intervention or control group (1:1) using the randomization module in REDCap, which allowed for randomization in real time to minimize the need for subsequent phone calls to the participants. The participants were stratified by B/AA or H/L race to maintain the balance between groups and ensure one group was not overrepresented in the sample. The random allocation sequence was generated using a random numbers table by two students from the SRO not participating in the enrollment to ensure concealment. All the screening, enrollment, consenting, and group assignment activities were conducted by students from the SRO.

### 2.4. Interventions

#### 2.4.1. Intervention Group

The participants randomized into the intervention group received 12 weeks of weekly home-delivered MTGs that adhered to the Dietary Approach to Stop Hypertension Diet (DASH) eating plan (Figure 1) provided by the FPP [17,33]. The deliveries weighed 13 pounds on average and consisted of 8–10 types of fruits and vegetables plus shelf-stable items sourced from local growers or produce distributors (Duke Campus Farm, Durham, NC, USA; and Farmer Foodshare, Durham, NC, USA) and the nearest food bank (Food Bank of Central and Eastern North Carolina, Raleigh, NC, USA), respectively. The participants were contacted before each home delivery to confirm availability to receive and store the MTG items.

The intervention group participants were also invited to attend up to six in-person nutrition education sessions, offered biweekly on Saturday afternoons at a central location in Durham. During the week leading up to each session, the participants were sent text and call reminders to encourage attendance and assess transportation needs. The study team members from the FPP arranged ride shares for the participants who indicated transportation barriers and provided on-site childcare during sessions to support equity in the study participation. Separate sessions were taught in both English and Spanish by language-concordant dietetics students from NCCU’s Dietetics Program with faculty preceptor oversight. The topics covered in each session adhered to the AHA EmPOWERED to Serve Health Lesson guidelines (Figure 2) [34], tailored to each racial/ethnic group by the dietetic student delivering the presentation. Each session included cooking demonstrations that introduced heart-healthy, culturally tailored recipes and taught the participants culinary skills for incorporating fresh produce into dishes. At the conclusion of each session, the participants were compensated with a USD 25 gift card to a local grocery store and received their MTGs delivery on site, which was tailored to include foods specifically discussed during the session.

The nutrition education sessions also doubled as data collection points, which allowed study team personnel from the FPP and bilingual CHWs to obtain anthropometric data for each participant (height, weight, and BP) [35] and administer surveys that assessed changes in household size, 24 h recall of fruit and vegetable consumption (FVC), antihypertensive use, and FI (USDA Short-Form Food Insecurity Scale, questions HH3-4 and AD1-3, amended to reflect the previous month at baseline assessment and previous two weeks at subsequent assessments) [36]. We chose to use the USDA questions at these sessions instead of the 2-question Hunger Vital Signs questionnaire used at the screening to permit the calculation of FI scores that may be more sensitive to changes in FI across the study period.

The participants were also screened for unmet health-related resource needs using the North Carolina Department of Health and Human Services’ social determinants of health screening questions [37] and offered assistance with any unmet needs identified. The participants were considered eligible to receive weekly home-delivered MTGs only after attending their first nutrition education session to capture the baseline anthropometric and survey data.

#### 2.4.2. Control Group

The participants randomized into the control group were invited to three in-person data collection sessions at the beginning, midpoint, and end of the 12-week study period. The study team members arranged ride shares for the participants that indicated transportation barriers for these sessions. At the conclusion of each session, the participants who attended were compensated with a USD 25 gift card to a local grocery store and a bag of MTG items equivalent to that provided to the intervention group that week.

#### 2.4.3. Home Visits

Two weeks before the end of the study period, we extended invitations to the participants in both groups who had completed two or fewer in-person sessions to participate in home visits. Home visits were coordinated by study team personnel and a language-concordant CHW to collect the survey and anthropometric data, and the participants remained eligible to receive compensation.

#### 2.4.4. Post-Intervention Period

To enhance the long-term equity for the entire study population, the participants from both groups were invited to enroll in the FPP at the conclusion of the study period to continue to receive biweekly home deliveries of free, locally sourced, fresh produce items. We also conducted an in-person, post-study dissemination session, where the study team members met with the study participants over lunch to share the preliminary results, offer opportunities to receive additional community resources, and administer a study experience feedback survey with quantitative and qualitative elements (Appendix A). The study team personnel from the FPP contacted the participants who did not attend the post-study dissemination session by phone to administer the study experience feedback survey and offer enrollment in the FPP.

In addition to the participant feedback, feedback from three individuals employed by the CBO partner (including the CBO Community Health Department manager, a study CHW, and the CHW’s direct coordinator) was obtained via a planned study reflection meeting that occurred after the post-study dissemination session. Two study team members (E.L. and S.B.) participated in this discussion and recorded notes.

### 2.5. Outcomes and Analytical Methods

To assess the intervention’s feasibility and acceptability (the study objectives), we drew from the Equity-Focused Implementation Research for Health Programs’ definitions of both terms to identify suitable primary outcomes [28]. For feasibility, defined from an equity-focused lens as the extent to which the intervention reduces barriers and can be carried out among disadvantaged populations, our prespecified primary outcomes were the eligibility rates associated with the screening criteria, enrollment rates, and retention rates across the study timepoints. To assess the acceptability, defined as the perception amongst the key players in implementation of the intervention, our prespecified primary outcomes were the post-study feedback from the participants and CBO partners. The secondary outcomes included changes in the BP, FVC, body mass index (BMI), and FI, and were chosen to align with outcomes commonly studied in similar FIM interventions and allow for comparison with extant FIM research [2].

#### 2.5.1. Statistical Methods

The participant demographics and characteristics were examined with descriptive statistics to assess for differences between the study groups using unpaired *t*-tests with alpha set at 0.05. To calculate the eligibility rates and enrollment rates, we documented the number of patients we identified in the EHR who met our prespecified eligibility criteria, the number of patients reached by phone to confirm eligibility, the number of patients eligible, and the number of patients who agreed to enrollment and randomization. We recorded the attendance at each study session and participation in the home visits to gauge the retention. The quantitative participant feedback from the post-study survey was examined using simple descriptive statistics.

For the secondary outcomes, only the participants who attended a minimum of two study sessions or home visits spaced at least two months apart were considered eligible for inclusion in the pre–post analyses to adequately capture changes over time (Appendix A). The FI scores were tabulated using the method described in the USDA U.S. Household Food Security Survey Module. The medians (Q1, Q3) were reported for ordinal data (FI) and discrete data (FVC) and the means (standard deviations) were reported for continuous data (BP and BMI) at the baseline and post-intervention [38]. The baseline differences between the intervention and control groups were assessed using unpaired *t*-tests with alpha set at 0.05. The pre–post changes in the secondary outcomes were expressed with 95% confidence intervals (CIs). Due to the nature of this feasibility and acceptability study, it was decided not to conduct efficacy statistical tests on any of the secondary outcomes.

#### 2.5.2. Qualitative Analytical Methods

To assess the qualitative participant feedback, a conventional content analysis was conducted by three independent reviewers (T.S., E.Y., and R.K.) to identify the emergent themes [39]. The initial phase involved immersion in the text through repeated readings to develop a general understanding, followed by the extraction of key concepts and meanings to form the initial codes. Subsequently, the codes were iteratively adjusted as their relationships, similarities, and differences were explored, and these were finally organized into clusters and categories while identifying unedited exemplars for each code. Disagreement between reviewers was resolved through discussion.

Additionally, the study team members who participated in the study reflection meeting with CBO representatives extracted themes from the meeting notes and compared them to the qualitative participant feedback.

## 3. Results

### 3.1. Participant Recruitment

Recruitment was conducted by 14 study team members and occurred during a two-week period between March and April 2023. A total of 1577 patients that met the prescreening eligibility criteria were identified in the EHR and the study personnel attempted to contact 265 to confirm eligibility. Of those contacted, 94 were successfully reached, 77 were screened as eligible, and 50 consented for randomization to either the intervention or control arm (25 each), which represented estimated post-screen eligibility and enrollment rates of 81.9% and 64.9%, respectively. The inability to make contact was the leading reason for non-enrollment (64.5%) (Figure 3).

### 3.2. Participant Demographics

The intervention group included 13 B/AA and 12 H/L participants, of whom 56% were female (53.8% in the B/AA group, 58.3% in the H/L group). The median (Q1, Q3) age was 55 (48, 62), and 32% reported Spanish as their preferred language. The control group included 12 B/AA and 13 H/L participants, of whom 68% were female (75% in the B/AA group, 61.5% in the H/L group). The median (Q1, Q3) age was 57 (49, 63), and 48% reported Spanish as their preferred language (Table 1).

### 3.3. Participant Retention

Amongst the intervention group, 68 out of 150 possible visits were completed (43%), which comprised 63 in-person nutrition education sessions visits (92.6%) and five home visits (7.4%). The participation waned over time, where fourteen participants attended at least half of the sessions (56%) and seven did not attend any sessions (28%). In the control group, 57 of 75 (76%) possible data collection visits were completed, which comprised 42 in-person data collection sessions (73.7%) and 15 home visits (26.3%). Fifteen participants completed all three visits (60%), five completed two visits (20%), two completed one visit (8%), and three (12%) did not attend any visits (Figure 4). Across both the study arms, four participants were provided with transportation to at least one study visit at their request and one participant made consistent use of the available childcare. The pilot trial ended as planned after the final nutrition education and data collection sessions. Seven participants were subsequently enrolled in the FPP to continue receiving biweekly produce deliveries.

### 3.4. Quantitative and Qualitative Participant Feedback

Eight intervention group participants and one control group participant attended the post-study dissemination session, with all nine completing the study experience feedback survey used to gauge acceptability. One additional intervention group participant and five control group participants completed the survey asynchronously. Of these 15 total respondents, ten self-reported as B/AA and five as H/L. The quantitative feedback showed that the respondents were highly satisfied (86.7%) or somewhat satisfied (13.3%) with the food provided during this study. Over half of the participants (53.3%) reported that the food closely matched the kinds of foods they liked to eat, and the remainder (46.7%) said the food somewhat closely matched the foods they liked to eat. All nine participants from the intervention group who completed the survey reported high satisfaction with the nutrition education classes. On a scale from 1–10, the average (standard deviation) overall satisfaction with the program was rated as 9.2 ± 0.9.

The conventional content analysis of the qualitative feedback from the participants that completed the survey revealed two main “Satisfaction” and “Dissatisfaction” clusters, which comprised six unique categories, namely, food provisions, knowledge, supports, agency, study logistics, and general endorsement. Within this framework, 25 unique codes were identified (Appendix A). The participant feedback was largely positive, and highlighted satisfaction with the quality and utility of both the food provisions and educational sessions. Multiple participants expressed appreciation for the healthiness and diversity of the provided produce and noted the improved cooking skills fostered by the study participation. Participants commented on the effectiveness of the study personnel communication and teaching at the nutrition education sessions. Moreover, the feedback highlighted important individual and community benefits conferred by the interventions, including enhanced food access and knowledge-sharing with the non-participants. Some participants pointed out instances where the food items received were not fresh, misaligned with their preferences, or unable to be prepared.

### 3.5. Qualitative CBO Feedback

The feedback provided by the CBO on the intervention’s acceptability and feasibility was generally positive. The CBO leaders commented on the intervention’s relevance to the highly food- and nutrition-insecure Latinx community they served, which was perceived to enhance the CHW engagement with the study team and strengthen the H/L participants’ interest. The CBO reported the execution of study activities assigned to CHWs was logistically uncomplicated and could be feasibly accomplished by two individuals. The CBO also noted smooth collaboration with the SRO leaders over the course of the study onboarding and intervention period.

The CBO also reported a few logistical barriers. The institutional research ethics trainings required for the CHW participation were not immediately available in Spanish and required translation, which delayed the study start. Additionally, due to scheduling constraints, the CBO noted that Saturday intervention sessions were more difficult to fulfill with CHW hours and reported it was easier to schedule CHWs to attend home visits during the week.

Finally, the CBO commented on areas for improvement in future studies. The CHWs stated that being better informed about education session content would enhance the impact of their interactions with participants. They also noted that reliance on the Duke Health EHR for recruitment could be potentially exclusionary for Durham H/L community members not receiving healthcare from Duke Health, and suggested recruitment from other local health systems in future studies.

Overall, the CBO agreed that the intervention was both acceptable and feasible for them and the H/L community they served.

### 3.6. Analysis of Secondary Outcomes

Fifteen participants in the control group and thirteen participants in the intervention group met the criteria for inclusion in the secondary outcome analyses. The groups were similar at baseline along all the secondary outcomes reported except for BMI (*p* = 0.02), with a statistically higher mean baseline BMI measured in the control group. The intervention was associated with absolute systolic and diastolic BP changes of −14.2 mmHg (CI −27.5, −4.5) and −9.5 mmHg (CI −17.6, −1.8), respectively, over the study period. The effect was more modest in the control group, which was associated with absolute systolic and diastolic BP changes of −3.5 mmHg (CI −11.7, −5.9) and 1.6 mmHg (CI −3.9, 7.5), respectively. Six intervention group participants (46%) and four (27%) control group participants experienced systolic BP reductions ≥10 mmHg. Five intervention group participants (38%) and four (27%) control group participants experienced diastolic BP reductions ≥10 mmHg. No changes in antihypertensive medication utilization were reported by participants in either group. Negligible changes in BMI were observed in either group (Appendix A).

The median FVC increased by four servings (CI 0.5, 4.0) in the intervention group and two servings (CI 1.5, 3.4) in the control group. Improvements in the FI were also reported in both groups, where five intervention group participants (38%) and five control group participants (33%) experienced a ≥2-point absolute reduction in the FI score. Median FI score changes of −2 (CI −2.2, −0.5) and −1 (CI −1.3, −0.2) were observed in the intervention and control groups, respectively.

### 3.7. Harms

No harms or unintended effects were reported by those that participated in either arm of this study.

## 4. Discussion

This 12-week pilot RCT aimed to assess the acceptability and feasibility of an intersectoral MTGs and nutrition education intervention that united an SRO with local CBOs to address BP and FI in underserved groups. We enrolled participants from historically underserved groups and integrated elements that addressed interrelated HRSNs (such as food and transportation insecurity) to develop diet-related interventions with the dual purpose of improving health equity [2,3].

### 4.1. Interpretation of Primary Outcomes

To evaluate the recruitment feasibility for a future trial, we calculated the eligibility and enrollment rates associated with our inclusion/exclusion criteria and study design. Our prescreening criteria identified over 1500 potentially eligible patients from a single healthcare system, which was made possible by the increased attention given to HRSN screening as a population health management strategy at both the national and institutional levels [4,5]. As screening and consenting were performed almost exclusively by students on a volunteer basis over a two-week period that coincided with midterms, we decided it was reasonable to reduce our recruitment goal from 100 to 50 participants without significantly impacting our ability to assess the trial’s primary and secondary objectives. Roughly 20% of the prescreened potential participants were enrolled mostly due to challenges making contact by phone. However, amongst those contacted, the eligibility and enrollment rates were reasonably high. Cumulatively, these findings suggest that a future RCT utilizing this pilot’s eligibility criteria and recruiting exclusively from one health system’s EHR is numerically feasible amongst the population of interest and could reasonably expect to enroll at least 200 participants by contacting over 1000 patients but would require a longer recruitment period and/or greater resources to hire students or staff to conduct screening and enrollment.

We also sought to evaluate retention as a gauge of the intervention’s feasibility. Community-based trials that work with traditionally underserved populations frequently encounter significant challenges with study retention and compliance due to structural and participant-related factors, which may include a lack of culturally sensitive staff, financial difficulties, employment constraints, transportation barriers, and medical distrust [10,17,18,40]. Several steps were taken in the planning and execution of this pilot trial to reduce barriers to participation and enhance retention, including partnership with well-established and trusted CBOs, childcare support, transportation to and compensation for study sessions, home MTG deliveries, and bilingual administration of all study activities. When we encountered difficulty getting some participants to respond to text and call reminders about upcoming study sessions and regularly attending in-person activities, we pivoted to offering home visits two weeks before the end of the study period.

Cumulatively, these efforts appear to have had a positive effect on the study retention, where 13 out of 25 participants (52%) in the intervention group and 15 out of 25 participants (60%) in the control group met the prespecified retention criteria for inclusion in the secondary outcome analyses. Attrition in the intervention group was the most related to insufficient attendance at the study sessions (16%), nonattendance following the enrollment despite contact (12%), and relocation (8%), while attrition in the control group was most commonly related to insufficient attendance at the study sessions (28%) and loss of contact (12%). An initial focus on home visits may have improved the retention to facilitate adequate collect data but may have also diminished any impact from in-person nutrition education sessions. However, the attrition rates associated with this study do appear to be on par with comparable interventions implemented for historically underserved groups [10,17]. Analogous pilot FIM studies found similar retention and attrition rates for in-person components, though some studies with fewer enrollees experienced lower attrition rates [12,14]. We speculate that offering home visits throughout the study, incorporating virtual options for nutrition education sessions, and providing greater clarity of trial activities during enrollment may enhance a future trial’s equity and retention, thereby further strengthening the feasibility.

The post-study quantitative and qualitative feedback obtained from the participants and the CBO partners to gauge the acceptability highlighted several strengths of the intervention. The participants were highly satisfied with the MTGs provided by the FPP across multiple domains. They verbalized the acquisition of new knowledge because of participation, focusing on food preparation and cooking skills, disease management, and food/nutrition knowledge. Several acknowledged indirect benefits from participation, including relationship-building, knowledge-sharing, and the agency to experiment with food and diet diversification, while noting the intervention strengthened access to nutritious food. Importantly, participants expressed an appreciation for several of the steps taken to reduce barriers to participation, including home visits, direct-to-door food delivery, transportation to study sessions, and compensation for study session attendance, suggesting that the elements intended to enhance the feasibility improved the acceptability as well [10]. The participants rated the intervention highly and tended to agree with the CBO partners that this study was relevant and beneficial to the minority communities it aimed to serve.

The qualitative feedback also pointed to areas for improvement. Three participants noted instances where the MTGs were not fresh or were unfamiliar, highlighting the need for increased quality control measures within the FPP and stronger coordination with NCCU’s dietetics students to ensure the relevance of education sessions to the items provided. Additionally, the participants were nearly evenly split in rating the food they received as “closely matching” and “somewhat closely matching” the foods they liked to eat. This points to a gap in acceptability, which is reflected in an emerging body of research showing that inequities in diet quality and health outcomes amongst ethnic minority groups may be partially attributable to nutrition interventions that fail to meet these groups’ cultural preferences [10,41]. While the intervention’s education sessions emphasized culturally tailored recipes, all participants received the same MTG items regardless of race or ethnicity. Future work should aim to resolve this gap by paying greater attention to the cultural tailoring and co-creation of MTG items with the populations of interest in the study design phase [10,12,17,41]. Finally, the CBO partners also identified challenges accessing institutional training from the SRO’s affiliated academic center and a desire to improve the CHW familiarity with nutrition education topics. Despite these observations, the preponderance of participant and CBO partner feedback affirmed the overall acceptability of the intervention studied in this pilot RCT.

### 4.2. Interpretation of Secondary Outcomes

While this pilot trial was not powered for hypothesis testing, our secondary outcome results suggest that an intervention that combines MTG deliveries with nutrition education sessions may have the potential to effectively target the FVC, FI, and BP in historically underserved populations. We observed a rise in the median FVC for the members of both groups, which was more pronounced in the intervention group, consistent with the findings from other MTG trials [2,11]. Additionally, while the median FI scores improved for both groups, the effect was larger in the intervention group, in line with previous studies [2,3,11]. We initially made the decision to assess the FI questions with respect to the previous month at baseline and with respect to the previous two weeks at subsequent visits to align with the two-week interval between nutrition education sessions. In retrospect, this may have led to inflated baseline FI scores. To reduce this bias, we recommend future studies that assess the FI status using consistent time intervals. The BMI did not change notably in either group, as expected for a trial that lasted only twelve weeks. Finally, patients in the intervention group appeared to experience a larger reduction in systolic BP and diastolic BP than the control group, though, again, this was not assessed for significance. Our decision to provide the control group participants with compensatory MTG bags and grocery store gift cards at the data collection sessions could have confounded the impact of the intervention on each of these secondary outcomes, which might be mitigated by extending the trial duration. Previous FIM trials that studied hypertension varied in length, with interventions that ranged from several weeks to years [2,3,11]. Planning for a definitive MTG trial should weigh the potential dose–response benefits of an extended time horizon against the resource requirements and retention challenges of a longer study, especially considering SRO and CBO timing constraints.

### 4.3. Generalizability

Several observations from this intersectoral collaboration between an SRO and CBO may be generalizable to others seeking to organize studies that leverage the strengths of each. This pilot demonstrated that trained student volunteers can administer most of the aspects of an MTGs clinical trial, including screening, consenting, enrollment, participant communications, data management, and delivery of elements of the intervention with support from faculty and CBOs. Findings also supported the capability of an SRO, like the FPP, to consistently source high-quality MTG items and reliably make home deliveries as part of a clinical trial. However, investigators should acknowledge that students may experience academic constraints that may limit their contributions to the academic year, which, along with the CBO CHW onboarding, factored into our decision to reduce the study duration from 24 weeks to 12 weeks when the regulatory approval delayed our study’s intended start date. Students’ unpredictable schedules can also lead to unanticipated turnover, as we experienced during the recruitment phase of this pilot trial. Additionally, CBOs contribute important strengths that enable SROs and research institutions to more fully engage with community members. The CBO representatives played a significant role in the participant-facing aspects of this study, which allowed for language-concordant support, greater study coverage, and increased racial and ethnic representation on the study team.

Study teams conducting powered MTG trials should consider engaging early and deeply with local CBOs and SROs for planning and leveraging their representatives to enhance the reliability, build participant trust, and ensure adequate personnel for the proper execution of the study protocol and data collection. This is especially vital for studies that incorporate important but labor-intensive elements that address HRSNs (like home deliveries and home visits) to promote health equity and study retention. Our findings reflect the activities of only one pilot trial; however, we believe that the methods—particularly our approach to addressing co-existing HRSNs—may serve as a template for novel pilot studies utilizing MTGs and other FIM interventions to improve health outcomes in minority and historically underserved populations.

### 4.4. Limitations

There are several limitations to this pilot study. First, as noted by the CBO CHWs in the post-intervention feedback, limiting enrollment to patients identified in the EHR of a single academic health center may exclude important segments of underserved communities and introduce selection bias toward a better-resourced population, for whom the intervention may be more feasible and acceptable [42]. This approach may also limit the generalizability of our findings to other healthcare systems with different patient demographics and community-level factors. Second, although the study personnel took care to assess the BP according to standardized clinical guidelines, measurements may have been influenced by several external stress-inducing factors beyond the research team’s control [43]. This might be mitigated by utilizing home BP assessment, as others have incorporated in study designs targeting blood pressure [44]. Third, the validity of the survey data on self-reported health and MTGs consumption may be influenced by the Hawthorne effect/social desirability bias or related response biases, which disproportionately affects minoritized groups [45,46]. Finally, we acknowledge that qualitative feedback from participants represents opinions from only those who remained engaged until the end of this study, which may bias our assessment of acceptability [47]. Similarly, while the participants in the intervention and control groups who met the criteria for inclusion in the secondary outcome analyses were statistically similar at baseline on most demographic and secondary outcome measures, attrition may have impacted the generalizability of our findings to the broader B/AA and H/L populations whose experiences we sought to understand.

### 4.5. Implications Toward a Definitive Trial

Despite these limitations, the results of this trial demonstrate that an SRO-delivered MTGs intervention paired with nutrition education sessions and undertaken in partnership with CBOs is feasible, acceptable, and has the potential to improve the health outcomes among underserved communities. To our knowledge, this was the first MTGs intervention with an explicit focus on B/AA and H/L communities, as well as the first RCT, pilot or otherwise, to investigate the acceptability and feasibility of using MTGs to treat hypertension [2,4]. We also show that including provisions that address coexisting HRSNs (such as home visits, transportation to in-person sessions, and home MTGs delivery) while designing FIM trials involving historically underserved groups may enhance the acceptability and feasibility. Based on the study objectives, there is clear promise in incorporating findings from this pilot trial toward a definitive and powered trial.

## Figures and Tables

**Figure 1 healthcare-13-00253-f001:**
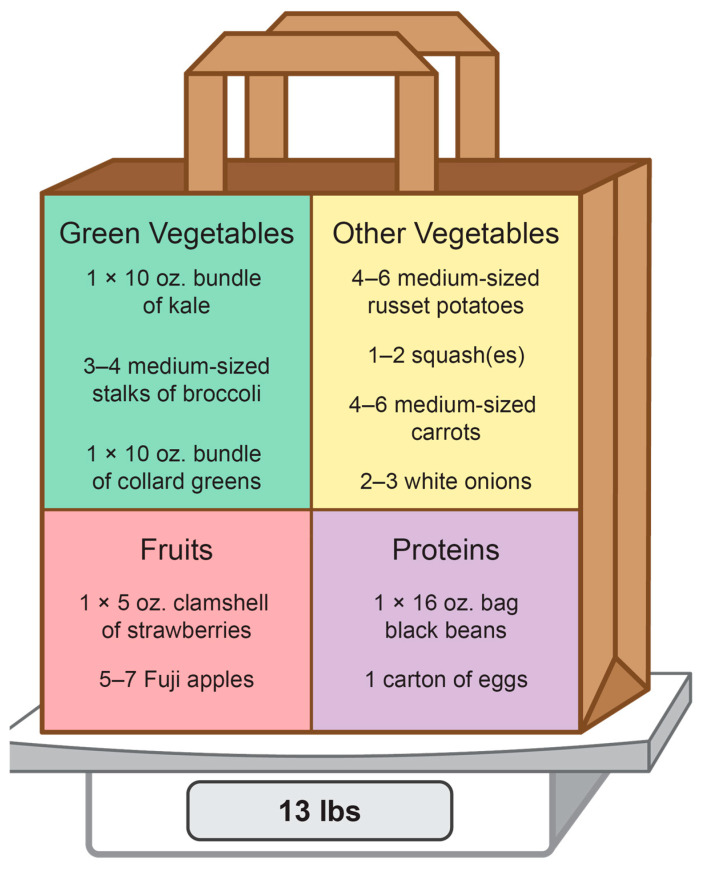
Example of one week of home-delivered medically tailored grocery items that adhered to the Dietary Approach to Stop Hypertension Diet. Deliveries were provided weekly to the participants in the intervention group.

**Figure 2 healthcare-13-00253-f002:**
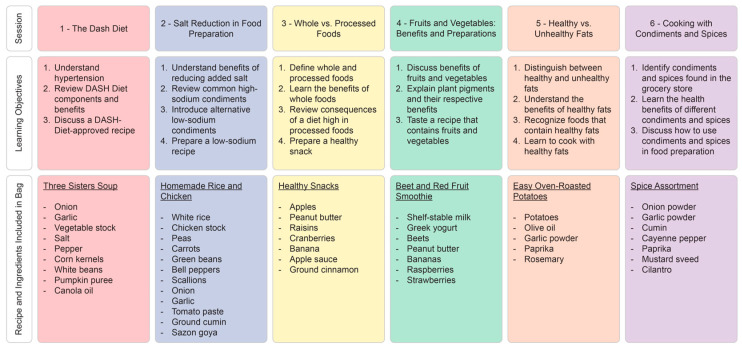
Learning objectives for in-person nutrition education sessions provided to the participants in the intervention group.

**Figure 3 healthcare-13-00253-f003:**
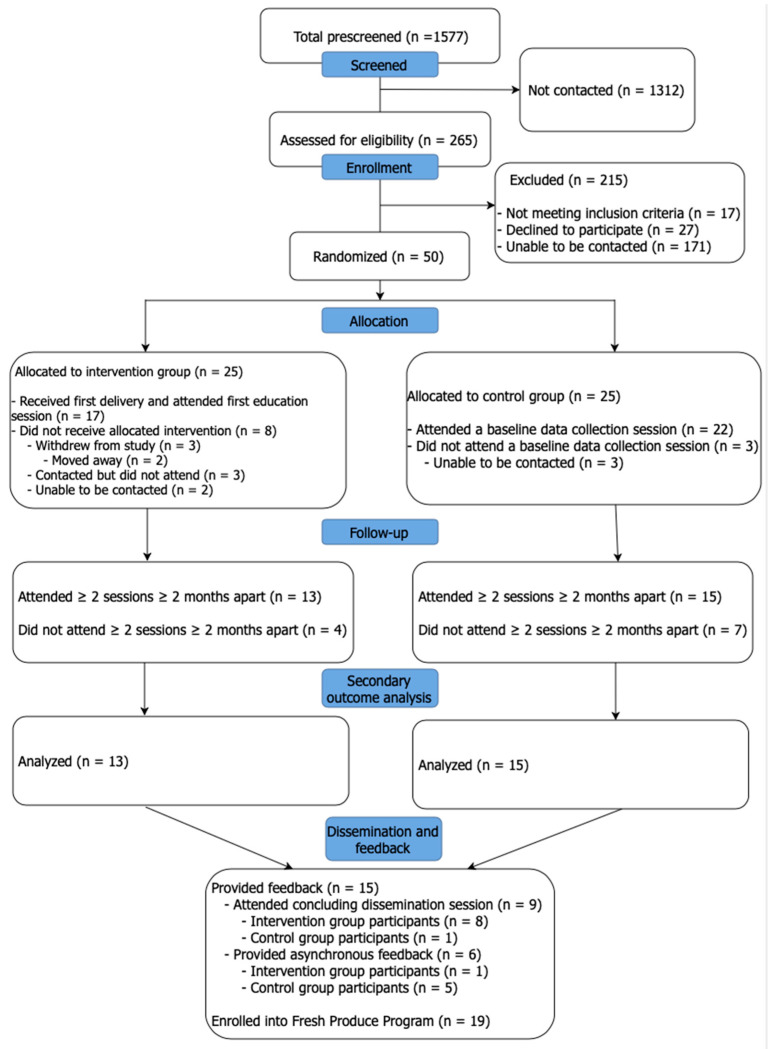
CONSORT diagram of participant flow.

**Figure 4 healthcare-13-00253-f004:**
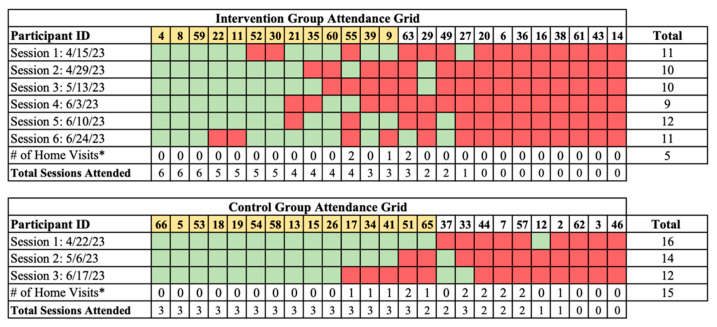
Study visit attendance. Green—present at session. Red—absent from session. Yellow—included in secondary analyses. The participants in the intervention group received weekly home MTG deliveries following their first session, regardless of the attendance at subsequent sessions. * Home visits were conducted between 21 June 2023 and 29 June 2023.

**Table 1 healthcare-13-00253-t001:** Demographics of study population. IQR—interquartile range.

Variable	Intervention (n = 25)	Control (n = 25)	*p*-Value
Age (years)			
Median (Q1, Q3)	55 (48, 62)	57 (49, 63)	0.75
Range	40–73	38–73	
Gender (%)			0.56
Female	14 (56)	17 (68)	
Male	11 (44)	8 (32)	
Race/ethnicity (%)			1.00
Black	13 (52)	12 (48)	
Hispanic/Latinx	12 (48)	13 (52)	
Primary language (%)			0.39
English	17 (68)	13 (52)	
Spanish	8 (32)	12 (48)	

## Data Availability

The data that support the findings of this study are available upon request from the corresponding author E.L. The data are not publicly available due to them containing information that could compromise the privacy of the research participants.

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
