# Peer review of "Medically Tailored Grocery Deliveries to Improve Food Security and Hypertension in Underserved Groups: A Student-Run Pilot Randomized Controlled Trial"

_healthcare, 2025, doi:10.3390/healthcare13030253_

Round 1
Reviewer 1 Report
Comments and Suggestions for Authors
Please see attachment

Reviewer 2 Report
Comments and Suggestions for Authors
Methods
Clarity and structure:
Rationale for design choices: explain why specific methods (e.g., using the REDCap randomization module or stratification by race/ethnicity) were chosen.
Participant recruitment: The process for identifying eligible participants through the EHR could be described more clearly.
Acknowledge potential limitations or biases introduced by relying on a single healthcare system for recruitment.
Intervention details: Highlight what makes the MTG and nutrition education sessions unique or innovative compared to other FIM programs.
Provide more context about why a 12-week duration was selected.
Outcomes:
Clearly distinguish between feasibility and acceptability as primary outcomes.
Include a brief justification for why secondary outcomes (e.g., BP and FI scores) were assessed despite the pilot's primary focus on feasibility and acceptability.
Results
The CONSORT diagram and participant flow could be summarized in text to complement the figure.
Quantitative detail:
Emphasize the primary outcomes (feasibility and acceptability) more clearly in the narrative before discussing secondary outcomes like BP and FI.
Ensure clarity when describing numerical results (e.g., absolute vs. relative changes).
Qualitative feedback:
Provide a clearer synthesis of participant and CBO feedback. Highlight recurring themes (e.g., logistical barriers or positive aspects of the intervention) more explicitly.
Limitations:
The discussion of limitations is well-structured, and the acknowledgment of potential selection bias due to reliance on the Duke Health EHR system is an important point. However, the handling of participant attrition could be further strengthened. While you appropriately highlight the attrition rates and mention the comparison of baseline characteristics between completers and non-completers, the analysis of attrition’s potential impact on outcomes could benefit from more depth.
Round 2
Reviewer 1 Report
Comments and Suggestions for Authors
Kudos on work well done!!